# Perforin 1 in Cancer: Mechanisms, Therapy, and Outlook

**DOI:** 10.3390/biom14080910

**Published:** 2024-07-26

**Authors:** Xiaoya Guan, Huina Guo, Yujia Guo, Qi Han, Zhongxun Li, Chunming Zhang

**Affiliations:** 1Shanxi Key Laboratory of Otorhinolaryngology Head and Neck Cancer, First Hospital of Shanxi Medical University, Taiyuan 030001, China; guohuina@sxent.org (H.G.); guoyujia@sxent.org (Y.G.); hanqi798@gmail.com (Q.H.); zhongxunli2016@163.com (Z.L.); 2Shanxi Province Clinical Medical Research Center for Precision Medicine of Head and Neck Cancer, First Hospital of Shanxi Medical University, Taiyuan 030001, China; 3Department of Otolaryngology Head & Neck Surgery, First Hospital of Shanxi Medical University, Taiyuan 030001, China

**Keywords:** PRF1, cancer, therapeutic targeting, prognostic, immune

## Abstract

PRF1 (perforin 1) is a key cytotoxic molecule that plays a crucial role in the killing function of natural killer (NK) cells and cytotoxic T lymphocytes (CTLs). Recent studies have focused on PRF1’s role in cancer development, progression, and prognosis. Studies have shown that aberrant PRF1 expression has a significant role to play in cancer development and progression. In some cancers, high expression of the PRF1 gene is associated with a better prognosis for patients, possibly because it helps enhance the body’s immune response to tumors. However, some studies have also shown that the absence of PRF1 may make it easier for tumors to evade the body’s immune surveillance, thus affecting patient survival. Furthermore, recent studies have explored therapeutic strategies based on PRF1, such as enhancing the ability of immune cells to kill cancer cells by boosting PRF1 activity. In addition, they have improved the efficacy of immunotherapy by modulating its expression to enhance the effectiveness of the treatment. Based on these findings, PRF1 may be a valuable biomarker both for the treatment of cancer and for its prognosis in the future. To conclude, PRF1 has an important biological function and has clinical potential for the treatment of cancer, which indicates that it deserves more research and development in the future.

## 1. Introduction

There is no doubt that cancer is one of the most dreaded diseases in the world, and its incidence is expected to continue to rise in the near future due to a variety of factors, which include our lifestyle, the environment, and the increase in life expectancy. Therefore, it is crucial to have a clear understanding of what cancer is. Cancer is an abnormal growth of cells that can occur in any structure or organ within the body. It has been suggested that the transformation of normal cells into cancer cells is not the important link, but rather that the newly formed cancer cells are too few in number to be recognized and destroyed by the body’s immune system [1]. With the increasing understanding of cancer, including its classification, pathogenesis, prevention, and treatment, the possibility of curing cancer has increased. As a matter of fact, the earlier cancer is detected, the higher the possibility of curing it [2].

As a member of the perforin/granzyme family, PRF1 is mainly secreted by natural killer cells and cytotoxic T lymphocytes. The perforin protein consists of 573 amino acid residues [3]. The PRF1 molecule has a molecular weight of about 60 kDa, and it consists of a transmembrane region, a hydrophobic region, and a perforin structural domain [3]. Upon contact of CTL with tumor cells, the transmembrane region of PRF1 is responsible for inserting it into the cell membrane, while the N-terminal hydrophobic region is responsible for the formation of pores in the tumor cell membrane, leading to tumor cell lysis and death [3].

The activity of PRF1 is influenced by a variety of factors, such as calcium ion concentration, pH, and reactive oxygen species concentration [4]. PRF1 also plays a crucial role in the immune response by forming immune synapses with virally infected or cancerous target cells and releasing them into the synaptic gap [4]. In order for cytotoxic lymphocytes and pro-apoptotic serine proteases, granzymes, to enter the cytoplasm of target cells and trigger apoptosis, PRF1 is required to form membrane pores. This process is critical for cancer treatment and prognosis [4].

As a perforating protein, PRF1 penetrates the cell membrane of target cells and triggers apoptosis [5]. This has always been the direction of commitment in cancer treatment [5]. In tumor cells and non-tumor cells, PRF1 acts under physiological as well as pathological conditions, and increasing evidence suggests that PRF1 secretion can form pores in tumor cell membranes, resulting in tumor cell lysis and death [5].

In this study, we introduce the role of PRF1 in cancer by describing the biological functions of PRF1 in tumor cells, as well as the role of PRF1 in tumor immunity and the microenvironment, and the therapeutic role of PRF1 in tumors and the prognosis of PRF1 on tumors in these sections.

## 2. Biological Functions of PRF1

In cytotoxic cells, perforin, encoded by the PRF1 gene, is a pore-forming protein stored in secretory vesicles that is released upon contact with target cells. A cytotoxic response is facilitated by it in the case of natural killer cells, CD8+ T cells, γδ+ T cells, and regulatory T lymphocytes. Perforin/granzyme pathway disruption results in persistent inflammation, chronic antigen presentation, and the release of inflammatory mediators. In addition to being associated with ineffective immune responses to viral infections (Herpesviridae), lymphohistiocytosis is associated with inflammatory and autoimmune diseases, transplant rejection, neoplastic diseases, and chronic inflammation [6,7].

As a pore-forming toxin, perforin (PRF, encoded by the PRF1 gene) is stored in cytotoxic lymphocyte secretory granules. During the immune response, these “killer” cells form immune synapses with virally infected or cancerous target cells and release PRF1 and granzyme serine protease. It is believed that PRF1 forms membrane pores, which are essential for allowing pro-apoptotic serine proteases and granzymes from cytotoxic lymphocytes to enter the target cell’s cytoplasm, thereby initiating apoptosis [8]. In cancer therapy, PRF1 is thought to play a role in promoting apoptosis and inhibiting cancer cell growth and spread. To improve the therapeutic efficacy and survival rate of cancer patients, we can better understand the biological functions of PRF1.

## 3. PRF1 Deficiency Leads to Tumor Immune Escape and Tumor Growth and Invasion

A gene called PRF1 is responsible for regulating perforin, which is located on chromosome 10q22.1 and consists of three exons. Nevertheless, only exons 2 and 3 are translated into the perforin protein, which is responsible for lymphocyte granule-mediated cytotoxicity [9,10]. Perforin acts on target cells in two different ways. In the first case, effector cells release their cytotoxic granules. Perforin forms holes directly in the plasma membrane of the target cell as a result of its very close proximity [9,10]. In the second model, the effector cell releases its cytotoxic granules, which are then endocytosed by the target cell [9,10]. In the endosome, perforin forms a pore in the membrane, allowing the granzyme to enter the cytoplasm. Inflammatory conditions caused by persistent infections and/or defective homeostatic perforin-mediated killing of activated immune cells may contribute to cancer susceptibility [11]. In patients deficient in perforin, the immune system is unable to kill target cells, since cytotoxic particles released by the immune system cannot penetrate these cells [12]. The natural killer (NK) cell secretes a cytolytic protein called Perforin-1 (Prf1). As a result of blocked perforin protein synthesis, resting NK cells contain high concentrations of Prf1 mRNA and exhibit minimal cytotoxicity, suggesting the presence of unknown post-transcriptional regulatory mechanisms [13,14].

A number of mechanisms are involved in immune escape from tumors, including antigenic modulation, immune checkpoints, the tumor microenvironment, defects in antigen processing and presentation, and apoptotic evasion of tumor cells [15]. It is well known that helper T cells and cytotoxic T cells are present in the tumor microenvironment, and this may enable the host system to naturally eliminate these immunogenic tumors [15]. Nevertheless, deletion of PRF1 decreases the sensitivity of tumor cells to the immune system, thereby facilitating their escape to cytotoxic T lymphocytes (CTLs) [15]. The absence of PRF1 may activate immune checkpoint pathways, such as the PD-1/PD-L1 pathway, leading to inactivation of T cells, reduced proliferation, and increased apoptosis, thus impairing their ability to recognize and attack tumor cells [16]. As a result, tumor cells are able to evade recognition and attack by the immune system [16]. Furthermore, PRF1 deficiency may result in alterations to the tumor microenvironment, such as the upregulation of immunosuppressive factors such as TGF-β and IL-10. As a result, immune cells would be inhibited from being able to recognize and attack tumor cells, preventing them from being effective [17]. Pyroptosis is a form of programmed cell death activated by inflammatory vesicles and characterized by cell swelling, cell membrane rupture, and release of pro-inflammatory cytokines. PRF1 has the potential to induce pyroptosis, and several studies have suggested that PRF1 may induce pyroptosis through a variety of mechanisms, e.g., the pore formed by PRF1 may lead to intracellular loss of potassium ions and activation of the NLRP3 inflammatory vesicle, which in turn induces Scorch death [18]. In general, PRF1 deficiency may weaken the antitumor immune response, causing tumor cells to evade recognition by the immune system and avoid attack.

Notably, immune effector cells include lymphokine-activated killer cells, cytokine-induced killer (CIK) cells, tumor-infiltrating lymphocytes (TILs), natural killer (NK) cells, γδ T cells, cytotoxic T lymphocytes (CTLs), and chimeric antigen receptor T cells, with the CIK cells being a heterogeneous population of CD3+ CD56+ and CD3+ CD56-cytotoxic T cells that have antitumor activity against multiple tumor targets [19]. A variety of tumor targets are targeted by CD3+CD56-cytotoxic T cells. CIK cells exhibit cytotoxicity influenced by perforin (PRF1), granzyme B (GZMB), Fas ligands (FASLs), CD40 ligands (CD40Ls), and a number of cytokines, whereas most immune checkpoint molecules and inflammatory tumor-promoting factors are suppressed in CIK cells [20]. It has been demonstrated that TIL play an important role in pro-tumor inflammatory and anti-cancer immune surveillance, and are also involved in the tumor microenvironment. It has been demonstrated that PRF1, as a key cytolytic effector, binds to CD8+ T cells and alters the clinical response to immune checkpoint inhibitors (ICIs). In order to determine the extent of immune infiltration, four immunogenes are usually used, CD274, CD8A, GZMA, and PRF1, which is usually represented by four immunogenes. Researchers have demonstrated that N6 methyladenosine (m6A) regulators may play an important role in the phenotypic modification of immune-related genes (IRGs), thus altering the characteristics of the tumor microenvironment (TME) by controlling TILs [21,22].

## 4. Role of PRF1 in Different Cancers

According to our analysis of the TCGA and TCGA+GTEX databases, PRF1 expression was higher in cancerous tissues than in paracancerous tissues. Additionally, PRF1 expression in immune cells was found to be relatively high in macrophages and T lymphocytes (Figure 1, Figure 2 and Figure 3). Hemophagocytic lymphohistiocytosis (HLH), for example, is a severe hyperinflammatory syndrome that includes methemoglobinemia, coagulation dysfunction, hepatic dysfunction, and hematopenia. There is a unifying pathophysiology of excessive inflammation and unrelieved T-cell and macrophage activation, often accompanied by NK cell dysfunction, causing cytokine-mediated systemic toxicity [11]. Originally, mutations in the perforin (PRF1) gene were associated with phagocytic lymphohistiocytosis (HLH), an inflammatory disorder [23,24]. The inheritance of double allele PRF1 mutations has been shown to result in fatal immunodeficiencies, familial hemophagocytic lymphohistiocytosis type 2 (FHL2), and other immunomodulatory disorders [25], including juvenile rheumatoid arthritis, macrophage activation syndrome, and chronic inflammatory demyelinating polyradiculoneuropathy [26]. It has been found that approximately 8–9% of Caucasian individuals carry the PRF1 mutation rs35947132 (c.272C>7), which results in the substitution of alanine for valine at residue 91 (p.A91V). Together with the null PRF1 allele, this mutation results in severe immunopathology [27,28]. A similar variant was found in childhood acute lymphoblastic leukemia (ALL) [29], but studies of adult ALL patients did not find the mutation, indicating that additional observations are required to explain the discrepancy. A study found four new single-allele missense mutations and two new single-allele synonymous mutations in PRF1 in 9 of 111 patients with all. All patients harboring PRF1 mutations were B-ALL with Ph chromosome or other cytogenetic abnormalities. It is demonstrated that mutations in PRF1 may play an important role in the pathogenesis of B-ALL [30].

During high-fat diet (HFD) feeding, PRF1 regulates T cell turnover, activation, and production of inflammatory cytokines in visceral adipose tissue (VAT) [31]. Since T cell activation in VAT regulates macrophage polarization and function, PRF1 appears to play an important role in obesity-associated chronic inflammation. In addition to its important role in cell-mediated immunity, PRF1 has been demonstrated to play a key role in immune-mediated disruption of the blood–brain barrier (BBB), which prevents peripheral immune cells, pathogens, and other bloodborne molecules from entering the central nervous system (CNS) [9]. It is believed that the blood–brain barrier is composed of a layer of brain endothelial cells adhering through tight junctions that isolate the central nervous system from the vascular system, and that the primary effector cells that can disrupt this barrier are still unclear [9]. In light of the newly discovered noncytolytic properties of perforin, it is possible that this molecule may mediate BBB disruption through unconventional mechanisms [9].

The overexpression of PRF1 in T lymphocyte subsets may contribute to the pathogenesis of asthma. A higher level of PRF1 and granzyme B expression in NK cells is associated with enhanced cytotoxicity in patients with allergic asthma. In healthy CD8+ T cells, methylation of the PRF1 methylation-sensitive region (MSR) suppresses gene expression, while demethylation of this site increases gene expression [32,33]. Among cancer patients with stage III and IV ovarian and basal-like breast cancer, PRF1 expression levels correlate with the infiltration of CD8+ T cells, dendritic cells, and neutrophils [34,35,36]. PRF1 forms membrane pores that release granzymes, resulting in the cytolysis of target cells. Defects in immune surveillance such as NK cell maturation, decreased NK activity, and reduced cytotoxicity contribute to breast cancer metastasis. As IL (including IL-2, IL-15, and IL-18) increases with time in culture, NK cell viability increases. Furthermore, IL (including IL-2, IL-15, and IL-18)-induced NK cells inhibited the proliferation of MCF7 cells (NK cells on breast cancer cells) and increased the release of TNF-α, IFN-γ, PRF1, and GzmB [37]. In pancreatic cancer, it has been reported that TIL interaction with tumor cells via chemokines (CCL4, CCL5, CXCL9, and CXCL10) triggers an exogenous apoptosis initiated by CD8+ effector T cells (regulation via PRF1, cell membrane perforation, enzymatic digestion via serine proteases, and exogenous apoptosis via ligand binding) [38]. A significant increase in HCC proliferation can be attributed to myeloid-derived suppressor cells (MDSCs), which reduce the number of CD8+ and CD4+ T cells, and one study found that MDSC and PRF1 mRNA expression are positively correlated in patients with liver tumors. Upon examining the data in more detail, it was found that a higher MDSC was associated with a lower PRF1 when observed at the T2 stage [39].

An immune-related genetic analysis was performed on 125 cold-frozen rectal tumor specimens in order to examine the balance between cytotoxic T cells and different helper T cell subpopulations in human colorectal cancer, as well as their impact on disease-free survival [40]. With the hierarchical clustering of correlation matrices, the functional clustering of genes associated with Th17 (RORC, IL17A), Th2 (IL4, IL5, IL13), Th1 (Tbet, IRF1, IL12Rb2, STAT4), and cytotoxicity (GNLY, GZMB, PRF1) was determined [40]. High levels of Th17 cluster expression were associated with a worse prognosis, while high levels of Th1 cluster expression were associated with a longer disease-free survival. Relapses were more easily distinguished when cytotoxicity/Th1 and Th17 clusters were analyzed together [40]. In conclusion, we found that PRF1 plays a very important role in a variety of cancers, and it is responsible for tumorigenesis, development, and metastasis, in addition to regulating cell cycle progression and modulating the tumor microenvironment. Further in-depth studies and reviews are needed on the contradictory findings of PRF1 in cancer progression. Some studies may support a positive role of PRF1 in cancer progression, while others may draw opposite conclusions. This contradiction may stem from different experimental conditions, differences in study subjects, or methodological limitations. Therefore, more studies are needed to clarify the exact mechanism of PRF1’s role in cancer progression in order to better guide the development of relevant therapeutic strategies. Therefore, an in-depth study of the function and mechanism of PRF1 will provide new ideas and strategies for cancer prevention, diagnosis, and treatment.

## 5. Cancer Treatment Modalities

Early detection and early diagnosis of cancer are extremely important as one of the most challenging health problems facing humans. Traditional treatments include surgery, radiotherapy, and chemotherapy. However, these treatments are only effective for early- and intermediate-stage cancers without lymphatic metastasis, and advanced cancer patients have a high mortality rate, so gene therapy has been further suggested in order to improve survival rates. Gene therapy has been proposed as a means of improving the survival rate of patients. The following are the advantages and disadvantages of various treatment methods.

### 5.1. Conventional Cancer Treatments

Conventional cancer treatments often combine surgery, radiotherapy, and chemotherapy to reduce recurrence risk [41,42,43].

Chemotherapy targets rapidly dividing cells, but causes common toxic side effects like hair loss and organ impairment. Complementary therapies such as acupuncture can manage these effects [41]. As a result of technological advances and the development of adjuvant chemotherapy, the outcomes of radiotherapy for cancer patients have been disappointing compared to the low treatment rates and inevitable side effects of radiotherapy. In contrast, other recent technologies such as adjuvant chemotherapy have shown better results in cancer treatment [43,44].

### 5.2. Emerging Technologies for Cancer Treatment

Emerging cancer therapies have made significant progress in addressing the limitations of traditional cancer treatments by exploring multiple dimensions, including innovative strategies such as gene therapy, immunotherapy, and gene transfer [45]. Immunotherapy reactivates the immune system to combat tumors, particularly aggressive ones, and has evolved to precision treatments like immune checkpoint inhibitors [46,47]. Adoptive cell therapy (ACT) uses engineered T cells (CAR-T) to target tumors [41,48,49]. Oncolytic viruses induce cell death and enhance T cell responses against tumors [50]. Personalized cancer vaccines targeting neoantigens are proposed for individualized treatment. Monoclonal antibodies target tumor antigens, inducing immune responses and cell death [23,41,51,52,53,54]. Gene transfer introduces new genes to control cell growth or induce cell death in tumors [45,55].

## 6. The Role of PRF1 in Cancer Therapy

Perforin 1 (PRF1) is a protein secreted by cytotoxic T lymphocytes (CTLs) and natural killer (NK) cells, which plays a key role in the immune system’s ability to kill tumor cells. Immunotherapy aims to activate or enhance immune cells so that they can recognize and destroy tumor cells more effectively. As PRF1 is a key factor in the killing of target cells by CTLs and NK cells, increasing the activity of these cells could directly enhance PRF1-mediated cytotoxicity. It systematically examined the key molecular targets of network pharmacology in cancer therapy and revealed the mechanism of action and targeting pathways of several anticancer drugs. As a result, this provides important theoretical guidance and support for our research, especially concerning PRF1 cancer immunotherapy [56]. For instance, cytokines such as interleukin-2 (IL-2) can increase the effectiveness of NK cells and CTLs against tumor cells by amplifying and activating their functions [37,57]. Checkpoint inhibitors, such as PD-1/PD-L1 and CTLA-4 inhibitors, can reverse the suppressive effects of the tumor on the immune system, restoring immune cell activity [58,59,60,61]. In prostate cancer research, PRF1-expressing liposomes driven by the PSA promoter were found to be effective in inducing perforin expression, thereby significantly inhibiting cancer cell growth. This strategy is expected to be an innovative approach for the treatment of advanced prostate cancer [62]. These treatments can indirectly increase the role of PRF1 because they enable CTLs and NK cells to attack tumor cells more aggressively, thereby increasing PRF1-mediated killing [63]. Adoptive cell therapies, such as chimeric antigen receptor T cell therapy (CAR-T), involve reprogramming the patient’s T cells in order to enable them to recognize and attack tumors. On contact with tumor cells, these modified T cells release PRF1, which causes them to die. It has been demonstrated that CAR-T therapy significantly enhances PRF1-mediated antitumor effects by increasing the specificity and persistence of T cells [64] (Figure 4).

## 7. The Role of PRF1 in Cancer Prognosis

The role of PRF1 in cancer prognosis is complex and depends on a number of factors [65], including the type of cancer, PRF1 expression levels, the tumor microenvironment, and the overall health of the patient. PRF1 expression is generally associated with a better prognosis in certain cancers, including melanoma, lung, breast, and colorectal cancers, where it indicates a strong anti-tumor immune response, which may inhibit tumor growth and spread. A variety of cancers, including head and neck cancer, esophageal cancer, and gastric cancer, are associated with poor prognosis when PRF1 expression is reduced or lost. Loss or reduction in PRF1 expression may weaken the anti-tumor immune response, which may make it easier for tumor cells to grow and spread. Although PRF1 remains controversial in terms of its role in cancer prognosis, there are a number of theories. Studies suggest that PRF1 activity may be influenced by the tumor microenvironment, such as inhibitory cytokines produced by tumor cells. PRF1 may therefore play a significant role in cancer prognosis depending on several factors, such as the type of tumor, its stage, the treatment regimen, and the patient’s overall health.

The lymph nodes that drain the tumor (TDLNs) play a significant role in the metastasis of most solid tumors which spread through the lymphatic system. A favorable prognosis for colorectal cancer (CRC) is associated with T cell infiltration, and the presence of CD8+, CD45RO+, and CD68+ T cells in TDLNs is associated with longer survival times. However, Foxp3+ Treg cells have been associated with immunosuppression and disease progression. In tumors with high expression of IFNG, PRF1, and GZMB, CD8+CXCR5+ T cells are associated with a longer disease-free survival, and their frequency has been shown to predict a better prognosis [66].

The Lewis lung cancer mouse model was used in one study to study the influence of gut flora on tumor treatment. As a consequence of impaired gut flora, cisplatin-conjugated antibiotic-treated mice developed enlarged tumors and had decreased survival rates. This was because ABX treatment decreased IFN-γ, GZMB, and PRF1 expression in CD8+ T cells, suggesting that gut flora contributes to an enhanced immune response. In contrast, mice treated with cisplatin in combination with Lactobacillus showed smaller tumors and higher survival rates because Lactobacillus increased the expression of IFN-γ, GZMB, and PRF1, which enhanced the antitumor effect of cisplatin [67].

In HCV-infected human hepatocellular carcinoma cells, PRF1 encodes perforin, which plays an important role in the lysis of CD8+ T cells [68,69]. Several studies have demonstrated that PRF1 plays a significant role in determining the prognosis of hepatocellular carcinoma (HCC) and predicting disease recurrence, especially after liver transplantation. According to the results of RNA-seq and real-time quantitative PCR, patients with recurrence have elevated PRF1 expression. In addition, five immune-related genes (IRGs)-FABP6, CD4, PRF1, EREG, and COLEC10-associated with the tumor microenvironment (TME) were significantly predictive of recurrence-free survival (RFS) and overall survival (OS) in patients with hepatocellular carcinoma. A key factor in lysogenic cell death, PRF1, has been identified as an important target in the treatment of hepatocellular carcinoma. The perforin-granzyme effector pathway is utilized by its related therapies to promote anti-tumor cytotoxicity [68,69,70].

PRF1 is a cytotoxic protein in CTL and NK cells that marks immune cells with killing capacity. In the tumor microenvironment (TME), the level of tumor-infiltrating lymphocytes (TILs) and immune status have a significant impact on tumor progression, treatment response, and recurrence. Head and neck squamous cell carcinoma (HNSCC) is an immunogenic and highly aggressive tumor, and an increase in TIL correlates with its favorable prognosis. Studies have shown that PRF1 expression correlates with immune infiltration in HNSCC, especially in HPV-positive HNSCC. The prognostic value of PRF1 is further supported by the fact that more CD8+ T cells, CD4+ T cells, NK cells, and M1-type macrophages have been observed in PRF1 highly expressed tumors [71,72].

Radical radiotherapy (CRT) is a less invasive treatment for esophageal squamous cell carcinoma (ESCC). By comparing pre- and post-treatment gene expression profiles, researchers found that 999 genes were overexpressed in complete remission (CR) cases, including at least 234 genes related to CTL activation in tumor-specific cytotoxic T lymphocytes. Among them, nine representative genes (PRF1, GZMB, IFNG, CASP1, CASP4, TNFSF4, TNFSE11, TNFSF13B, and TNFAIP8) were significantly overexpressed in the post-treatment samples, especially in CR cases. These results suggest that CTL is activated by CRT in CR cases and may provide a prognostic advantage for ESCC with epithelial features [73].

Overall, PRF1’s role in cancer prognosis is complex and depends on multiple factors. We analyzed the correlation of PRF1 in cancer prognosis through the TCGA database and found that the prognosis was poorer in renal papillary cell carcinoma (KIRP), low-grade glioma of the brain (LGG), and ovaral melanoma (UVM), while it was better in adrenocortical carcinoma (ACC), breast invasive carcinoma (BRCA), head and neck carcinoma (HNSC), sarcoma (SARC), skin melanoma (SKCM), and endometrioid carcinoma (UCEC) (Figure 5). But further studies are needed to elucidate the exact role of PRF1 in different cancer types and to determine the potential for its use as a prognostic biomarker.

## 8. Conclusions and Outlook

PRF1 is a perforator protein that plays an important role in the body’s immune defenses. A major part of its expression takes place within natural killer cells (NK cells) and cytotoxic T lymphocytes (CTLs), which are responsible for recognizing and destroying cells that are infected with viruses or tumors. PRF1 plays an important role in various aspects of tumor cell development, immune escape mechanisms, immunotherapy for cancer, as well as prognosis. It is important to understand that PRF1’s mechanism of action has a significant impact on the prognosis of cancer when it comes to treatment. The immune system can recognize and destroy tumor cells, improving patients’ survival chances. Some cancer patients have low levels of PRF1 expression in their immune systems, which may contribute to tumor growth and spread. PRF1 also plays an important role in cancer immune escape. Immune escape is the process by which tumor cells evade recognition and attack by the immune system through various pathways, which leads to cancer growth and spread. Understanding the mechanism of PRF1’s role in immune escape can help scientists better understand cancer development and provide an important reference for the development of new therapeutic approaches.

In summary, PRF1’s impact on cancer encompasses multiple aspects, such as biological function, treatment, and prognosis, and its mechanism of action is critical for improving cancer treatment and monitoring. An in-depth study of PRF1’s mechanism of action may provide an important reference for the treatment and prognosis of cancer, as well as provide increased hope for patient survival.

## Figures and Tables

**Figure 1 biomolecules-14-00910-f001:**
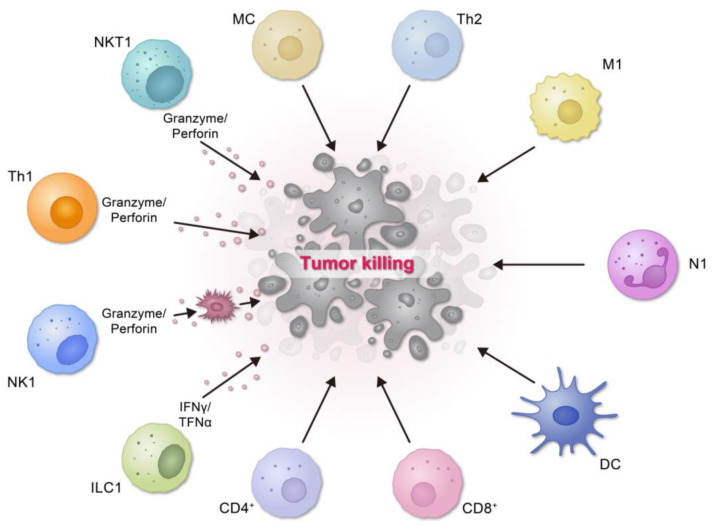
Perforin 1 in Cancer: Immune Function, Tumor Targeting, and Cytolysis.

**Figure 2 biomolecules-14-00910-f002:**
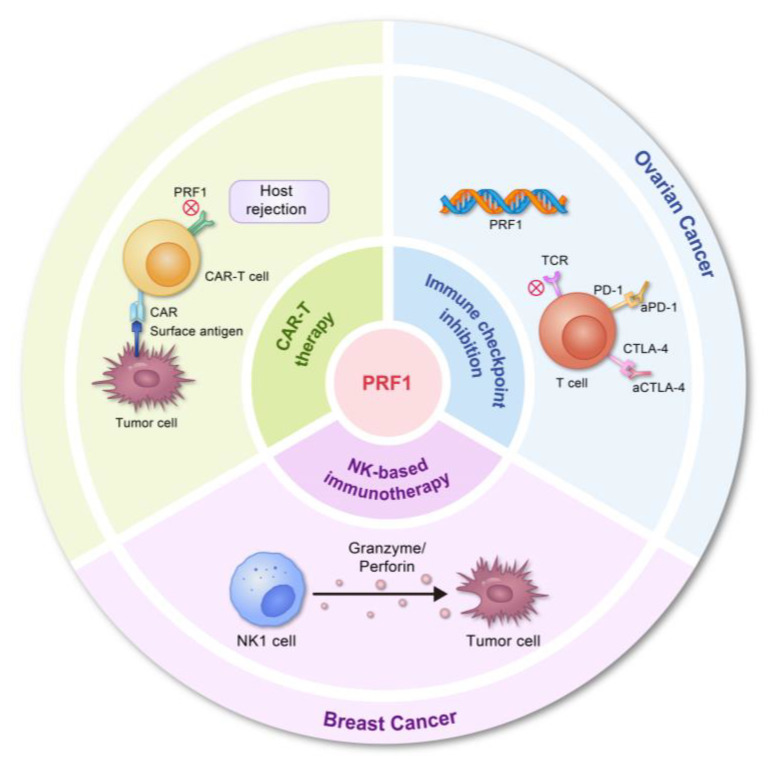
Perforin 1 in Cancer: Therapy Modalities Across Cancer Types.

**Figure 3 biomolecules-14-00910-f003:**
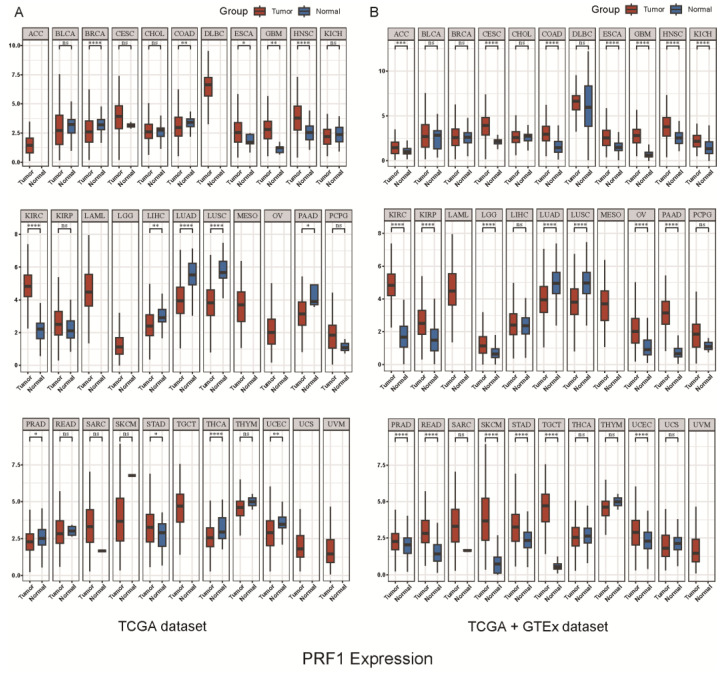
(**A**,**B**) TCGA dataset/TCGA+GTEx dataset: Distribution of PRF1 gene expression in tumor tissues and normal tissues, where the horizontal coordinates represent different tumor tissues and the vertical coordinates represent the distribution of the expression of this gene, where different colors represent different groups, * *p* < 0.05, ** *p* < 0.01, *** *p* < 0.001, **** *p* < 0.0001, pand asterisks represent the degree of significance (* *p*). The significance of the two groups of samples was tested using a Wilcox test.

**Figure 4 biomolecules-14-00910-f004:**
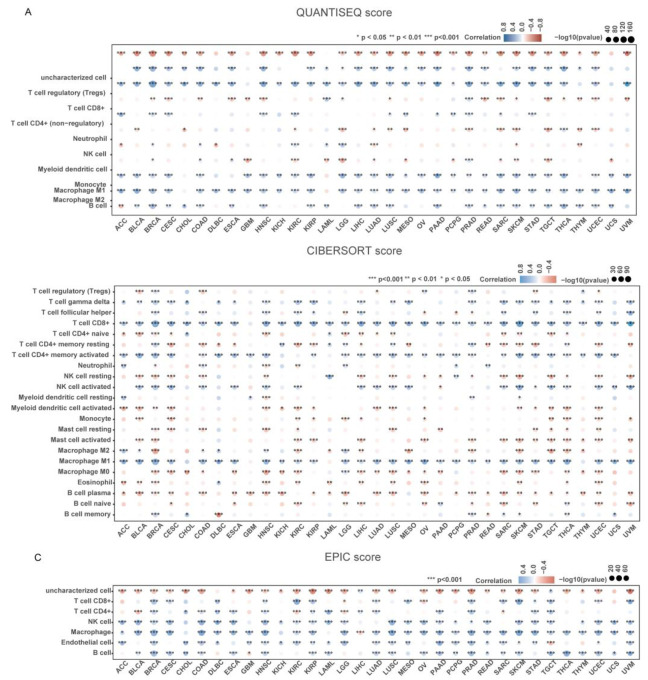
(**A**–**C**) Heatmap of correlation analysis between QUANTISEQ/CIBERSORT/EPIC immune infiltration score and Spearman’s correlation analysis of PRF1 gene expression in multiple tumor tissues, where the horizontal coordinates represent different tumor tissues, the vertical coordinates represent different immune infiltration scores, the different colors represent the correlation coefficients, the negative value represents negative correlation, the positive value represents positive correlation, and the stronger the correlation the darker the color, * *p* < 0.05, ** *p* < 0.01, *** *p* < 0.001, and asterisks represent significance (* *p*).

**Figure 5 biomolecules-14-00910-f005:**
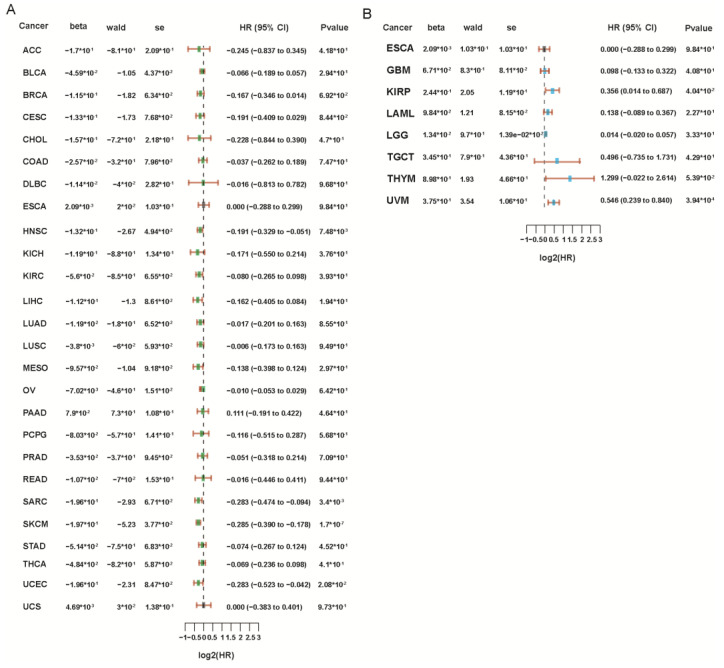
(**A**,**B**) Forest plot, results of one-way cox analysis of PRF1 gene in multiple tumors, *p*-value, risk coefficient of HR, confidence interval of HR, β-value, Wald value, and SE.

## Data Availability

Not applicable.

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
