# Peer review of "Perforin 1 in Cancer: Mechanisms, Therapy, and Outlook"

_biomolecules, 2024, doi:10.3390/biom14080910_

Round 1

Reviewer 1 Report

Comments and Suggestions for Authors

This paper is a well-written review of the various functions that the well-known Perforin 1 protein can exhibit in relation to tumors. Researchers studying Perforin 1 protein are expected to gain valuable information from this work.

1. It seems necessary to increase the resolution of Figures 4 and 5 to make them clearer.

2. It is necessary to clarify the citations for each sentence. There appear to be many sentences without citations, so this needs to be reviewed.

Author Response

Comments 1: It seems necessary to increase the resolution of Figures 4 and 5 to make them clearer.

Response 1: Thank you for pointing this out. I/We agree with this comment. Therefore, we have optimized Figures 4 and 5 to improve their clarity. The optimized images have been uploaded to the system so that readers can view and understand our research results more clearly. Once again, we thank the experts for their suggestions and we will keep trying to improve them in order to enhance the quality and readability of the paper.

Comments 2: It is necessary to clarify the citations for each sentence. There appear to be many sentences without citations, so this needs to be reviewed.

Response 2: Agree. We have, accordingly, revised some of the literature to emphasize this point. Here are the details:

Line 42-49, cite this article "Law, R., Lukoyanova, N., Voskoboinik, I. et al. The structural basis for membrane binding and pore formation by lymphocyte perforin. Nature 468, 447–451 (2010). https://doi.org/10.1038/nature09518"

Line 50-56, cite this article "Voskoboinik I, Dunstone MA, Baran K, Whisstock JC, Trapani JA. Perforin: structure, function, and role in human immunopathology. Immunol Rev. 2010 May;235(1):35-54. doi: 10.1111/j.0105-2896.2010.00896.x. PMID: 20536554. "

Line 57-61, cite this article" Metkar, S., Marchioretto, M., Antonini, V. et al. Perforin oligomers form arcs in cellular membranes: a locus for intracellular delivery of granzymes. Cell Death Differ 22, 74–85 (2015). https://doi.org/10.1038/cdd.2014.110"

Line 85-92, cite this article" [9,10] "

Line 102-109, cite this article" [15] "

Line 109-113, cite this article" [16] "

Line 182-192, cite this article" [9] "

Line 216-225, cite this article" [40] "

Reviewer 2 Report

Comments and Suggestions for Authors

The authors have presented a concise commentary on the role of perforin 1 in the cancer while emphasizing on the mechanism of its action. The authors also commented on its role in cancer prognosis. However, there are some concerns regarding the article which are as below:

1.     The language has can be improved by edit from a native speaker. There are various phrases and sentences in the article which are ambiguous-

a.     Line 16, “devcancer” should be “cancer”.

b.     Line 16, “e cancers..” should be “In cancers…”.

c.     Line 18 is contradictory to line 17 as from these two sentences it seems high expression of PRF1 can lead to better prognosis as well as immune escape of the tumor.

d.     Line 245, what is “ipation”.

e.     Line 256, “Radiation therapy is oneor a wide….” should be “Radiation therapy is used in a wide….”

f.       Line 259, “better control of their diseaand bettered to other treatment options.” Should be framed as “better control of their disease.”

g.     Line 263, “As a result……radiation therapy [38,40].” is ambiguous. I assume authors intend to compare the radiation therapy with adjuvant chemotherapy and other recent technologies, but the sentence used in the article doesn’t indicate so.

2.     Line 156, the name of the mutation is “c.272C>7” instead of “(c.272C4T)”.

3.     In line 160, authors mention total of six mutations and then indicate that of these 7 in BCR-ABL gene which doesn’t add up. I would suggest authors to verify and re-write this part.

4.     Line 190, the term “constant” should be removed.

5.     Line 220, authors claim role of perforin in metastasis, regulation of cell cycle progression and influencing the DNA damage repair. However, there is no support for these claims. Further, there has been attempt to verify the role of PRF1 in cancer, but the finding has been contradictory. I would suggest authors to also comment about that as the role of PRF1 in cancer progression or development still needs investigation.

6.     Authors discuss immunotherapy and monoclonal antibody as classification of Gene Therapy of cancer, which is not true. I would suggest authors condense the section 5 as discussion about treatment cancer treatment strategies seems out of scope of this review.

7.     In regard to the gene therapy, I would suggest authors to discuss about the article https://www.nature.com/articles/s41598-021-03324-6 (Gene therapy of prostate cancer using liposomes containing perforin expression vector driven by the promoter of prostate-specific antigen gene).

Comments on the Quality of English Language

Author Response

Comments 1: The language has can be improved by edit from a native speaker. There are various phrases

and sentences in the article which are ambiguous-

a. Line 16, "devcancer" should be "cancer".

b. Line 16, “e cancers.." should be "In cancers...".

c. Line 18 is contradictory to line 17 as from these two sentences it seems high

expression of PRF1 can lead to better prognosis as well as immune escape of the tumor.

d. Line 245, what is “ipation".

e. Line 256,"Radiation therapy is oneor a wide...." should be "Radiation therapy is

used in a wide...."

f. Line 259, "better control of their diseaand bettered to other treatment options."

Should be framed as "better control of their disease."

g. Line 263, "As a result......radiation therapy [38,40]." is ambiguous. I assume authors

intend to compare the radiation therapy with adjuvant chemotherapy and other recent technologies, but the sentence used in the article doesn't indicate so.

Response 1: Thank you for pointing this out. We agree with this comment. Therefore, we have made a few changes.Here is a detailed explanation.

a. Thank you to the experts for pointing out this issue. Line 16, PRF1 expression has a significant role to play in the cancer development and progression.

b. Thank you to the experts for pointing out this issue. Line 17, In cancers, high PRF1 expression has been associated with a better prognosis, possibly due to the fact 17 that it enhances the immune response to the tumor.

c. Thank you to the experts for pointing out this issue. Line17,"In cancers, high PRF1 expression has been associated with a better prognosis, possibly due to the fact that it enhances the immune response to the tumor. High PRF1 expression, on the other hand, may promote tumor escape and contribute negatively to patient survival." revise "In some cancers, high expression of the PRF1 gene is associated with a better prognosis for patients, possibly because it helps enhance the body's immune response to tumors. However, some studies have also shown that the absence of PRF1 may make it easier for tumors to evade the body's immune surveillance, thus affecting patient survival. ".

d. Thank you to the experts for pointing out this issue. Line 245, "ipation" means "diarrhea or constipation".

e. Thank you to the experts for pointing out this issue. Line 256,"Radiation therapy is oneor a wide...." revise "Radiation therapy is used in a wide range of cancers, including head and neck cancer, lung cancer, cervix cancer, bladder cancer, prostate cancer, and skin cancer.".

f. Thank you to the experts for pointing out this issue. Line 259, "better control of their diseaand bettered to other treatment options." revise" According to clinical studies, radiation therapy provides patients with a longer survival time and better control of their disease.".

g. Thank you to the experts for pointing out this issue. Line 263, "As a result......radiation therapy [38,40]." revise "As a result of technological advances and the development of adjuvant chemotherapy, the outcomes of radiotherapy for cancer patients have been disappointing compared to the low treatment rates and inevitable side effects of radiotherapy. In contrast, other recent technologies such as adjuvant chemotherapy have shown better results in cancer treatment [38,40].".

Comments 2: Line 156, the name of the mutation is“c.272C>7"instead of “(c.272C4T)".

Response 2: Thank you for pointing this out. We agree with this comment. Therefore, we have made a few changes.Line 159, "It has been found that approximately 8-9% of Caucasian individuals carry the PRF1 mutation rs35947132 155 (c.272C>7), which results in the substitution of alanine for valine at residue 91 (p.A91V). "

Comments 3: In line 160, authors mention total of six mutations and then indicate that of these 7 in BCR-ABL gene which doesn't add up. I would suggest authors to verify and re-write this part.

Response 3: Thank you for pointing this out. We agree with this comment. Therefore, we have made a few changes.In line 164, "The present study identified…B-ALL [27]. " revise "A study found four new single-allele missense mutations and two new single-allele synonymous mutations in PRF1 in 9 of 111 patients with ALL. All patients harboring PRF1 mutations were B-ALL with Ph chromosome or other cytogenetic abnormalities. It is demonstrated that mutations in PRF1 may play an important role in the pathogenesis of B-ALL [27]. ".

Comments 4: Line 190, the term“constant”should be removed.

Response 4: Thank you for pointing this out. We agree with this comment. Therefore, we have made a few changes. Line194, " A higher level of PRF1 and granzyme B expression in NK cells is associated with enhanced cytotoxicity in patients with allergic asthma. ".

Comments 5: Line 220, authors claim role of perforin in metastasis, regulation of cell cycle progression and influencing the DNA damage repair. However, there is no support for these claims.Further, there has been attempt to verify the role of PRF1 in cancer, but the finding has been contradictory. I would suggest authors to also comment about that as the role of PRF1 in cancer progression or development still needs investigation.

Response 5: Thank you for pointing this out. We agree with this comment. Therefore, we have made a few changes.We will also consider adding a comment on the contradictory findings of PRF1 in cancer development to show that this area still needs more exploration and research. We will endeavor to ensure the accuracy and completeness of the article.

Line 227, we will remove "influencing DNA damage repair". We will add "Further in-depth studies and reviews are needed on the contradictory findings of PRF1 in cancer progression. Some studies may support a positive role of PRF1 in cancer progression, while others may draw opposite conclusions. This contradiction may stem from different experimental conditions, differences in study subjects, or methodological limitations. Therefore, more studies are needed to clarify the exact mechanism of PRF1's role in cancer progression in order to better guide the development of relevant therapeutic strategies. "

Comments 6: Authors discuss immunotherapy and monoclonal antibody as classification of Gene Therapy of cancer, which is not true. I would suggest authors condense the section 5 as discussion about treatment cancer treatment strategies seems out of scope of this review.

Response 6: Thank you for pointing this out. We agree with this comment. Therefore, we have made a few changes."5.1. Conventional Cancer Treatments

Conventional cancer treatments often combine surgery, radiotherapy, and chemotherapy to reduce recurrence risk [41–43]. Chemotherapy targets rapidly dividing cells but causes common toxic side effects like hair loss and organ impairment. Complementary therapies such as acupuncture can manage these effects [38,40,41]. As a result of technological advances and the development of adjuvant chemotherapy, the outcomes of radiotherapy for cancer patients have been disappointing compared to the low treatment rates and inevitable side effects of radiotherapy. In contrast, other recent technologies such as adjuvant chemotherapy have shown better results in cancer treatment [43,44].

5.2. Emerging Technologies for Cancer Treatment:

Emerging cancer therapies have made significant progress in addressing the limitations of traditional cancer treatments by exploring multiple dimensions, including innovative strategies such as gene therapy, immunotherapy, and gene transfer [45]. Immunotherapy reactivates the immune system to combat tumors, particularly aggressive ones, and has evolved to precision treatments like immune checkpoint inhibitors [46,47]. Adoptive cell therapy (ACT) uses engineered T cells (CAR-T) to target tumors [41,48,49]. Oncolytic viruses induce cell death and enhance T cell responses against tumors [50]. Personalized cancer vaccines targeting neoantigens are proposed for individualized treatment. Monoclonal antibodies target tumor antigens, inducing immune responses and cell death [23,41,51–54]. Gene transfer introduces new genes to control cell growth or induce cell death in tumors [45,55]. "

Comments 7: In regard to the gene therapy, I would suggest authors to discuss about the article https://www.nature.com/articles/s41598-021-03324-6(Gene therapy of prostate cancer using liposomes containing perforin expression vector driven by the promoter of prostate- specific antigen gene).

Response 7: Thank you for pointing this out. We agree with this comment. Therefore, we have made a few changes.Line282, we will add "In prostate cancer research, PRF1-expressing liposomes driven by the PSA promoter were found to be effective in inducing perforin expression, thereby significantly inhibiting cancer cell growth. This strategy is expected to be an innovative approach for the treatment of advanced prostate cancer [62]. ".
